# Efficacy and Safety of Nivolumab Plus Ipilimumab for Metastatic Renal Cell Carcinoma in Patients 75 Years and Older: Multicenter Retrospective Study

**DOI:** 10.3390/cancers17030474

**Published:** 2025-01-31

**Authors:** Shimpei Yamashita, Shuzo Hamamoto, Junya Furukawa, Kazutoshi Fujita, Masayuki Takahashi, Makito Miyake, Noriyuki Ito, Hideto Iwamoto, Yasuo Kohjimoto, Isao Hara

**Affiliations:** 1Department of Urology, Wakayama Medical University, 811-1 Kimiidera, Wakayama 641-0012, Japan; ykohji@wakayama-med.ac.jp (Y.K.); hara@wakayama-med.ac.jp (I.H.); 2Department of Nephro-urology, Nagoya City University Graduate School of Medical Sciences, 1 Kawasumi, Mizuho-cho, Mizuho-ku, Nagoya 467-8601, Japan; hamamo10@med.nagoya-cu.ac.jp; 3Department of Urology, Kobe University Graduate School of Medicine, 7-5-1 Kusunoki-cho, Chuo-ku, Kobe 650-0017, Japan; jfurukawa@tokushima-u.ac.jp; 4Department of Urology, Kindai University Faculty of Medicine, 377-2 Onohigashi, Osakasayama 589-0014, Japan; kfujita@med.kindai.ac.jp; 5Department of Urology, Tokushima University Graduate School of Biomedical Sciences, 3-18-15 Kuramo-to-cho, Tokushima 770-8503, Japan; takahashi.masayuki@tokushima-u.ac.jp; 6Department of Urology, Nara Medical University, 840 Shinjo-cho, Kashihara 634-8521, Japan; makitomiyake@naramed-u.ac.jp; 7Department of Urology, Japanese Red Cross Wakayama Medical Center, 4-20 Komatsubara-dori, Wakayama 640-8558, Japan; ito-n.14@wakayama-med.jp; 8Division of Urology, Department of Surgery, School of Medicine, Faculty of Medicine, Tottori University, 86 Nishi-cho, Yonago 683-8503, Japan; gakkoura@tottori-u.ac.jp

**Keywords:** renal cell carcinoma, immunotherapy, nivolumab plus ipilimumab, elderly, efficacy, tolerability

## Abstract

Our multi-center retrospective cohort study assesses the efficacy and safety of nivolumab plus ipilimumab in patients ≥75 years old. A comparison of efficacy between patients ≥75 years (*n* = 33) and patients <75 years (*n* = 123) showed no significant differences in objective response rate, disease control rate, progression-free survival, or cancer-specific survival. However, overall survival in patients ≥75 years was significantly poorer than that in patients <75 years. Moreover, an age ≥75 years was shown in multivariable analysis to be a significant independent predictor of poorer overall survival. Toxicity was not significantly different between the two groups.

## 1. Introduction

The introduction of immune checkpoint inhibitors (ICIs) has revolutionized treatment strategies against metastatic renal cell carcinoma (mRCC). Several phase 3 randomized controlled trials have shown that ICI combination therapies have better therapeutic effects for previously untreated advanced clear cell renal cell carcinoma compared with sunitinib, a tyrosine kinase inhibitor that was previously the primary first-line therapy for mRCC [1,2,3,4,5]. These ICI combination therapies are therefore currently the dominant first-line therapy for mRCC, especially metastatic clear cell renal cell carcinoma.

The Checkmate 214 trial compared nivolumab plus ipilimumab (NIVO + IPI) with sunitinib. Survival and response rates were significantly higher with NIVO + IPI than with sunitinib among International Metastatic Renal Cell Carcinoma Database Consortium (IMDC) intermediate- and poor-risk patients with previously untreated advanced renal-cell carcinoma [1]. Moreover, extended 8-year follow up results showed superior survival, durable response benefits, and an acceptable safety profile maintained with NIVO + IPI compared with sunitinib at 8 years [6]. These long-term survival benefits and the durability of responses are major advantages of the NIVO + IPI combination therapy.

However, there are several concerns about the clinical benefits of NIVO + IPI in elderly patients with mRCC. Their expected life expectancy may be shorter than in younger patients, for example, because of aging and comorbidities. Also, the immune environment in elderly patients is different from that in younger patients, potentially affecting the efficacy and immune-related adverse event (irAE) incidence of NIVO + IPI [7]. Similarly, the treatments for irAEs can be more harmful in elderly patients than in younger patients [8]. The global median age at diagnosis of kidney cancer is reportedly approximately 75 years, so it is extremely important to accumulate real world data on the clinical outcomes of NIVO + IPI in elderly patients with mRCC [9].

Although the treatment outcomes of NIVO + IPI in elderly patients with mRCC have been the focus of several recent studies, there is still limited evidence for the efficacy and safety [10,11,12]. This multicenter retrospective study therefore aims to compare the clinical outcomes of NIVO + IPI between patients ≥75 years and patients <75 years and to evaluate the efficacy and safety of NIVO + IPI in patients ≥75 years with mRCC.

## 2. Patients and Methods

This multi-institutional retrospective study included the 156 consecutive patients who received NIVO + IPI treatment for mRCC between October 2015 and May 2022 at our hospital and seven collaborating hospitals. It was conducted in accordance with the Declaration of Helsinki after approval by the Institutional Review Boards of Wakayama Medical University (approval number 3470) and the other seven hospitals. Informed consent was deemed unnecessary, owing to the retrospective nature of this study.

The combination therapy of nivolumab 3 mg/kg and ipilimumab 1 mg/kg was received every three weeks for up to four cycles, and this was then followed by nivolumab monotherapy. These treatments were continued until disease progression or unacceptable toxicity. If NIVO + IPI treatment was discontinued, patients were treated according to the judgment of each physician.

We retrospectively examined the data on patient demographics and tumor characteristics at the start of NIVO + IPI treatment from medical records. This included age, sex, prior resection at the primary site, histologic subtype, IMDC risk classification, and the number and locations of metastatic sites. We also assessed the clinical outcomes of NIVO + IPI treatment, including the response of tumors to NIVO + IPI, progression-free survival (PFS), cancer-specific survival (CSS), overall survival (OS), and the toxicity of NIVO + IPI. The response of tumors was evaluated using Response Evaluation Criteria in Solid Tumors version 1.1 [13]. The objective response rate (ORR) was defined as the proportion of patients with complete response (CR) or partial response. The disease control rate (DCR) was defined as the proportion of patients with CR, partial response, or stable disease (SD). PFS was defined as the time from the initiation of NIVO + IPI treatment to either radiological and clinical disease progression or to death. CSS was defined as the time from the initiation of NIVO + IPI treatment to death from mRCC or to the patient’s last follow-up visit. OS was defined as the time from the initiation of NIVO + IPI treatment to death from any cause or to the patient’s last follow-up visit.

Based on the previous report that the global median age at the diagnosis of kidney cancer is 75 years, we classified the patients using 75 years as a cutoff [9] and compared the patient demographics, tumor characteristics, and clinical outcomes between patients ≥75 years and patients <75 years. The comparisons of patient demographics, tumor characteristics, and the toxicity of NIVO + IPI were performed using the chi-squared test or Fisher’s exact test. PFS, CSS, and OS rates were determined by the Kaplan–Meier method. Comparisons of PFS, CSS and OS between the two groups were performed using log rank tests. Univariable and multivariable Cox proportional regression analyses were performed to identify predictors of PFS, CSS, and OS. All statistical analyses were performed using JMP Pro 16 (SAS, Cary, NC, USA), and *p* < 0.05 was considered to be statistically significant in all analyses.

## 3. Results

### 3.1. Comparison of Patient Demographics and Tumor Characteristics

Among 156 patients, the histologic subtype was clear cell in 103 patients (66%), non-clear cell in 36 patients (23%) (papillary in 18 patients, unclassified in 11 patients, chromophobe in two patients, translocation in two patients, spindle cell in two patients, and collecting duct carcinoma in one patient) and unknown in 17 patients (11%), respectively. The median age in overall patients was 69 years (quartile; 60–74 years) and 33 and 123 patients were classified into patients ≥75 years old and patients <75 years old, respectively. A comparison of patient demographics and tumor characteristics between the two groups is shown in Table 1; there were no significant differences between the two groups except for age.

### 3.2. Comparison of Best Tumor Response to NIVO + IPI Treatment

A comparison of the best tumor response to NIVO + IPI treatment between the two groups is shown in Figure 1. The best tumor response in those ≥75 years old was CR in three patients (9%), partial response in 12 patients (36%), SD in six patients (18%), and progressive disease in 10 patients (30%). In those <75 years old, the best tumor response was CR in 13 patients (11%), partial response in 42 patients (34%), SD in 35 patients (28%), and progressive disease in 27 patients (22%). The ORR was 45% (15/33 patients) in patients ≥75 years old and 45% (55/123 patients) in the other group; there was no statistical difference in the ORR between the two groups (*p =* 0.93). The DCR was 64% (21/33 patients) in patients ≥75 years old and 73% in the other patients; there was also no statistical difference in the DCR between the two groups (*p =* 0.29).

### 3.3. Comparison of Survival Outcomes

Median PFS was six months [95% CI: 2–9] in patients ≥75 years old and 11 months [95% CI: 7–19] in the other patients; there was no significant difference in PFS between the two groups (*p =* 0.05) (Figure 2A). During the median follow-up duration of 21 months (quartile: 8–31 months) among the patients ≥75 years old, 14 died of mRCC (42%) and six died of other causes (18%) (pneumonia *n* = 3, cerebral hemorrhage *n* = 1, cardiovascular disease *n* = 1, and chronic kidney disease *n* = 1). Among the other patients, 42 died of mRCC (34%) and eight died of other causes (7%). The median CSS was 27 months [95% CI: 13—not reached] in patients ≥75 years old and 51 months [95% CI: 39—not reached] in the other patients; there was no significant difference in CSS between the groups (*p =* 0.10) (Figure 2B). The median OS was 18 months [95% CI: 9–33] in patients ≥75 years old and 46 months [95% CI: 29—not reached] in the other patients, so OS in the patients ≥75 years old was significantly shorter than in the other patients (*p =* 0.01) (Figure 2C).

### 3.4. Multivariable Analyses for PFS, CSS, and OS

Multivariable analyses of associations between predictive factors and PFS, CSS, and OS are shown in Table 2. Prior resection at the primary site was the only significant independent predictor of PFS (*p <* 0.01), and prior resection of the primary site and histologic subtype were significant independent predictors of CSS (*p <* 0.01 and *p <* 0.01, respectively). Meanwhile, in addition to prior resection at the primary site and the histologic subtype (*p <* 0.01 and *p <* 0.01, respectively), age ≥75 years was independently associated with poor OS (*p =* 0.01).

### 3.5. Comparison of Toxicity of NIVO + IPI

A comparison of irAEs between the two groups is shown in Table 3. There were no significant differences between them in the toxicity of NIVO + IPI (including severe irAEs, corticosteroid use for an irAE, and treatment discontinuation due to an irAE).

## 4. Discussion

In this multicenter retrospective cohort study, we compared the clinical outcomes of NIVO + IPI between patients ≥75 years old and patients <75 years old with mRCC. We also evaluated the efficacy and safety of NIVO + IPI in patients ≥75 years old. No significant differences in the ORR, PFS, CSS, or the toxicities of NIVO + IPI were found between the two age groups. However, OS in the patients ≥75 years old was significantly shorter than that in the other patients. Moreover, age ≥75 years old was shown to be independently associated with poor OS. The clinical outcomes of NIVO + IPI in elderly patients with mRCC has been the focus of several recent studies, but the cutoff value of age varies between the studies, and the number of elderly patients included in each study has been comparatively small. It is therefore important to accumulate real-world data on the clinical outcomes of NIVO + IPI in elderly patients with mRCC. We used a cutoff of 75 years because it represents the global median age at the diagnosis of kidney cancer [9]. In addition, this multicenter study is thought to be especially valuable because it is the largest to evaluate the efficacy and safety of NIVO + IPI in patients aged ≥75 years.

Various ICI combination regimens for mRCC have been evaluated in phase 3 randomized controlled trials in comparison with sunitinib, and they have been approved as first-line treatment regimens for patients with mRCC [1,2,3,4,5]. Recently, combination regimens of ICI and tyrosine kinase inhibitors, such as pembrolizumab plus axitinib, avelumab plus axitinib, nivolumab plus cabozantinib, and pembrolizumab plus lenvatinib, seem to be used more frequently due to their high response rates [14]. However, NIVO + IPI, one of the earliest ICI combination therapies, is still recommended by guidelines as the first-line treatment regimen in IMDC intermediate-/poor-risk patients with mRCC [15]. The main advantages of NIVO + IPI are considered to be the durability of response and long-term survival benefit, which might be a different characteristic from that of sunitinib because it has been reported that most of the advanced mRCCs treated with sunitinib develop intrinsic drug resistance [16]. On the other hand, extended follow-up results of the Checkmate 214 study recently demonstrated that NIVO + IPI maintained superior survival, a durable response, and a manageable safety profile vs. sunitinib at 8 years, which is the longest follow-up of any phase 3 trial of ICI combination therapy [6]. However, the efficacy of NIVO + IPI for elderly patients was not fully assessed by the results of the Checkmate 214 trial; the rate of patients ≥75 years was only 8.2% of the patients who received NIVO + IPI. In addition, a subgroup analysis of IMDC intermediate-/poor-risk patients showed no statistical difference in OS between the NIVO + IPI group and the sunitinib group in the patients aged ≥75 years (HR: 0.97, 95% CI: 0.48–1.95), while OS in NIVO + IPI group was superior to that in the sunitinib group in patients <65 years [1].

There are several concerns related to the immune system in elderly people. Immune senescence, immune system deterioration, and impaired adaptive and cell-mediated immunity secondary to thymic atrophy may be associated with poor survival in elderly people [17]. Immune senescence is characterized by a decrease in naive CD8^+^ T cells coupled with an increase in memory subsets, skewed T cell receptor repertoire, and immune cells producing a tolerogenic environment composed of regulatory T cell populations [18,19,20]. Memory T cells made from naive T cells in elderly people have lower proliferation rates and lower effector cytokine production compared with memory T cells made from naive T cells in younger people, which can lead to inferior immune responses [21,22]. Epigenetic changes in immune cells and cancer cells with aging might also be involved in the anticancer response [23,24].

The effects of aging on the immune checkpoint system are still unclear [25], but the efficacy and safety of NIVO + IPI for elderly patients with melanoma have been reported [26,27]. For example, Stoff et al. compared the PFS and toxicity of NIVO + IPI between 26 patients ≥75 years and 32 patients <75 years with melanoma and reported that there were no statistical differences in these outcomes between the groups [27]. Moreover, several recent studies have focused on the treatment outcomes of NIVO + IPI in patients with mRCC. Nemoto et al. retrospectively examined 60 patients who received NIVO + IPI for mRCC. There were no significant differences in ORR, DCR, PFS, OS, or in the incidence rate of irAEs between patients aged ≥70 years (*n* = 20) and those aged <70 years (*n* = 40) [10]. This suggests that NIVO + IPI has possible efficacy with tolerable toxicity in elderly patients, but the survival findings are controversial due to the short follow-up period (median follow-up period: 10.6 months). Elsewhere, Kobayashi et al. compared the treatment outcomes between patients ≥75 years old (*n* = 26) and the other patients (*n* = 94) who were administered NIVO + IPI for mRCC [11]. No intergroup differences were shown in the ORR, PFS, OS, or in the incidence of irAEs, and the age cutoff was the same as that used in our study. However, after propensity score matching with various covariates including histology, prior nephrectomy, clinical stage at first diagnosis, and IMDC risk classification, OS in the patients ≥75 years old tended to be shorter than that in the other patients (HR 1.703, 95% CI: 0.960–3.021, *p* = 0.069). They suggested that this can be due to natural causes.

In the present study, we also compared the clinical outcomes of NIVO + IPI between the patients aged ≥75 years and others, demonstrating that there were no significant differences in the ORR, DCR, PFS, CSS, or in the incidence of irAEs between two groups, which is consistent with the results of previous studies. On the other hand, OS in the patients ≥75 years was significantly shorter than that in the patients <75 years, so older age was an independent significant predictor of poor OS. No significant difference in CSS was found between the groups, so this difference in OS might be because of the death by natural causes in elderly patients, as suggested by Kobayashi et al. Our results suggest that the indication for NIVO + IPI in elderly patients should be carefully considered, taking into account each patient’s life expectancy. Comprehensive geriatric assessment has been demonstrated to have a strong prognostic value for survival and toxicity in elderly patients receiving chemotherapy for many advanced solid tumors [28,29]. In the field of treatment for mRCC, Pierantoni et al. classified elderly patients who received tyrosine kinase inhibitors for their mRCC into fit, vulnerable, or frail by using the Comprehensive Geriatric Assessment. The assessment could accurately discriminate patients with a higher risk of shorter survival and severe toxicities [30]. It may be appropriate to determine the indication for NIVO + IPI based not simply on age but also on various patient backgrounds, including the results of the Comprehensive Geriatric Assessment.

There are a number of limitations to the current study; it was a retrospective study and it involved a limited number of patients. To the best of our knowledge, however, this multicenter study is the largest to evaluate the clinical outcomes of NIVO + IPI in patients ≥75 years with mRCC. A further limitation is that the subjects of our study with mRCC were all Japanese patients, and the clinical outcomes of NIVO + IPI could be influenced by geographic region, mainly because the pharmacokinetics may differ depending on ethnicity [31]. Further large-scale studies in different regions are needed to verify our results.

## 5. Conclusions

The clinical efficacy of NIVO + IPI, including the ORR, PFS and CSS, in patients ≥75 years with mRCC was comparable with that in patients <75 years. The toxicity of NIVO + IPI in was also comparable between the groups. Meanwhile, OS in patients ≥75 years old was significantly shorter, and older age was an independent significant predictor of poor OS. The indication for NIVO + IPI in elderly patients should therefore be carefully considered, taking into account each patient’s life expectancy.

## Figures and Tables

**Figure 1 cancers-17-00474-f001:**
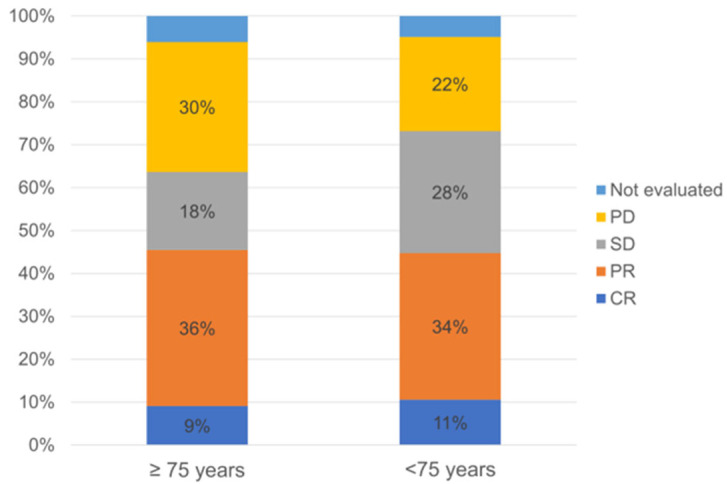
Comparison of the best tumor response between patients ≥75 years old and patients <75 years old. CR: complete response; PR: partial response; SD: stable disease; PD: progressive disease.

**Figure 2 cancers-17-00474-f002:**
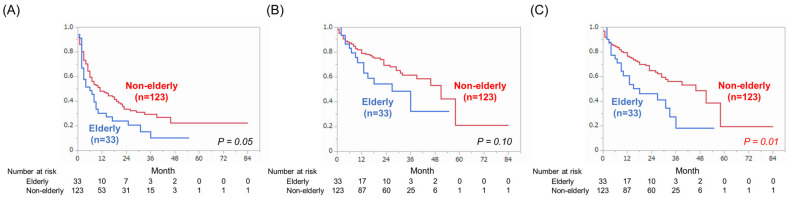
Comparison of (**A**) progression-free survival, (**B**) cancer-specific survival, and (**C**) overall survival between patients ≥75 years old and patients <75 years old.

**Table 1 cancers-17-00474-t001:** Comparison of patient characteristics between patients ≥75 years old and patients <75 years old.

	Patients ≥75 Years Old (*n* = 33)	Patients <75 Years Old (*n* = 123)	
Age, years	78 (77–83)	66 (58–70)	<0.01
Sex, *n* (%)			0.66
Male	26 (79)	101 (82)	
Female	7 (21)	22 (18)	
Prior resection at the primary site, *n* (%)			0.81
Yes	19 (58)	68 (55)	
No	14 (42)	55 (45)	
Histologic subtype, *n* (%)			0.94
Clear cell	22 (67)	81 (66)	
Non-clear cell	7 (21)	29 (24)	
Unknown	4 (12)	13 (11)	
IMDC risk classification, *n* (%)			0.17
Intermediate	24 (73)	74 (60)	
Poor	9 (27)	49 (40)	
Number of metastatic organs, *n* (%)			0.65
Single	16 (48)	65 (53)	
Multiple	17 (52)	58 (47)	
Metastatic sites, *n* (%)			
Lung	22 (67)	71 (58)	0.34
Lymph nodes	11 (33)	46 (37)	0.66
Bone	8 (24)	44 (36)	0.20
Liver	4 (12)	19 (15)	0.78
Adrenal gland	4 (12)	7 (6)	0.24
Brain	0 (0)	3 (2)	1.00
Others	10 (30)	22 (18)	0.12

Continuous variables are shown in “median (IQR)” form. IMDC: International Metastatic RCC Database Consortium.

**Table 2 cancers-17-00474-t002:** Multivariable analyses of factors associated with progression-free survival, cancer-specific survival, and overall survival between patients ≥75 years old and patients <75 years old.

	Progression-Free Survival			Cancer-Specific Survival			Overall Survival		
	HR	95% CI	*p* Value	HR	95% CI	*p* Value	HR	95% CI	*p* Value
Age: ≥75 years (vs. <75 years)	1.50	0.96–2.33	0.07	1.74	0.92–3.24	0.08	2.00	1.16–3.42	0.01
Prior resection of primary site: no (vs. yes)	1.81	1.21–2.70	<0.01	2.62	1.46–4.68	<0.01	2.45	1.46–4.10	<0.01
Histologic subtype: clear cell (vs. others)	0.83	0.55–1.24	0.37	0.42	0.24–0.73	<0.01	0.50	0.30–0.82	<0.01
IMDC risk classification: poor (vs. intermediate)	1.10	0.73–1.63	0.64	1.04	0.60–1.80	0.87	1.20	0.73–1.94	0.46
Number of metastatic organs: multiple (vs. single)	1.18	0.80–1.73	0.40	1.12	0.65–1.93	0.67	1.23	0.76–1.99	0.39

IMDC International Metastatic RCC Database Consortium.

**Table 3 cancers-17-00474-t003:** Comparison of immune-related adverse events between patients ≥75 years old and patients <75 years old.

	Patients ≥75 Years (*n* = 33)	Patients <75 Years (*n* = 123)	*p* Value
Any grade, *n* (%)	21 (64)	85 (69)	0.55
Grade ≥3, *n* (%)	12 (36)	47 (38)	0.84
Corticosteroid use for an irAE, *n* (%)	9 (27)	52 (42)	0.11
High-dose corticosteroid use for an irAE, *n* (%)	2 (6)	25 (20)	0.06
Treatment discontinuation due to an irAE, *n* (%)	10 (30)	38 (30)	0.98

irAE: immune-related adverse event.

## Data Availability

Derived data supporting the findings of this study are available from the corresponding author SY on request.

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
