# Peer review of "Efficacy and Safety of Nivolumab Plus Ipilimumab for Metastatic Renal Cell Carcinoma in Patients 75 Years and Older: Multicenter Retrospective Study"

_cancers, 2025, doi:10.3390/cancers17030474_

Round 1
Reviewer 1 Report
Comments and Suggestions for Authors
Thanks for the opportunity to review this interesting paper comparing the outcomes of elderly patients with metastatic renal cancer treated with nivolumab + ipilimumab with those of younger patients.
The study adds valuable information that may be useful to guide clinical decisions in these patients. It is the largest report of its kind, and the previous literature has been adequataly reported and discussed.
The manuscript is clear and well-written and the data is well.presented. The discussion is balanced and the limitations are commented and discussed.
My only minor comment is:
- Consider adding and discussing the results of the study with nivolumab + ipilimumab in elderly patients with melanoma (line 221, reference 226).
Author Response
Comments 1: Consider adding and discussing the results of the study with nivolumab + ipilimumab in elderly patients with melanoma (line 221, reference 26).
Response 1:Thank you for your valuable suggestion. We added the following sentence to the discussion section.
For example, Stoff et al. compared PFS and toxicity of NIVO+IPI between 26 patients ≥ 75 years and 32 patients < 75 years with melanoma, and reported that there were no statistical differences of these outcomes between the groups [27]. (lines 229-231)
Reviewer 2 Report
Comments and Suggestions for Authors
The authors focused on the efficacy aspects of treating advanced, metastatic RCC in 156 patients with nivolumab plus ipilimumab (NIVO+IPI). In the division over 75 years old, there was a significant decrease in median survival from 46 months vs. 18 months compared to younger patients. These are data that may change the treatment algorithm for mRCC in this age group, although they require studies by other teams on other patient populations.
RCC is a tumor that correlates with age. Moreover, the survival rate of metastatic cancer patients is also related to age. In this case, it would be appropriate to consider the sensibility of treatment with the combination of NIVO+IPI given the results presented by the authors.
The paper is valuable, I have a few comments on the corrections:
-what test/method and was the minimum number of patients in this study determined at all?
- why the division of patients into 2 groups with a cutoff at 75? It is not a split at a median age of 69 as in the results. OK - I found hidden in the discussion - please also write this at the beginning of the results.
- The low percentage (about 66%) of ccRCC compared to 75-80% of ccRCC in other countries is puzzling. What were the other histological subtypes of RCC?
- Have patients been investigated for VHL syndrome?
- quite a few patients died from non-mRCC related events (18% for >75 yo). What were the causes of death and could they be related to side effects of NIVO+IPI?
- In the discussion, I did not find a clear statement that discontinuation of treatment with sunitinib is associated with the acquisition of resistance to this chemotherapeutic and subsequent proliferation of tumor cells ? Could this be the case with NIVO+IPI ? The paper DOI: 10.3892/ijo.2019.4830 may be helpful in this case.
Overall, the results of the study are interesting, a good contribution to changing the treatment algorithm for mRCC.
Author Response
Comments 1: what test/method and was the minimum number of patients in this study determined at all?
Response 1: Thank you for your important question. We did not determine the minimum number of patients included in this study because this was an observational retrospective study.
Commnts 2: why the division of patients into 2 groups with a cutoff at 75? It is not a split at a median age of 69 as in the results. OK - I found hidden in the discussion - please also write this at the beginning of the results.
Response 2: We appreciate your valuable suggestion. We added the following sentence to the methods section before the beginning of the results.
Based on the previous report that a worldwide median age at diagnosis of kidney cancer is 75 years, we classified the patients using 75 years as a cutoff [9], and compared the patient demographics, tumor characteristics and clinical outcomes between patients ≥ 75 years and patients < 75 years. (lines 106-109)
Comments 3: The low percentage (about 66%) of ccRCC compared to 75-80% of ccRCC in other countries is puzzling. What were the other histological subtypes of RCC?
Response 3: Thank you for your important question. The histologic subtype of non-clear cell RCC was papillary RCC in 18 patients, unclassified RCC in 11 patients, chromophobe RCC in 2 patients, translocation RCC in 2 patients, spindle cell RCC in 2 patients and collecting duct carcinoma in 1 patient, respectively. We added the following sentence to the results section.
Among 156 patients, the histologic subtype was clear cell in 103 patients (66%), non-clear cell in 36 patients (23%) (papillary in 18 patients, unclassified in 11 patients, chromophobe in 2 patients, translocation in 2 patients, spindle cell in 2 patients and collecting duct carcinoma in 1 patient) and unknown in 17 patients (11%), respectively. (lines 118-121)
Comments 4: Have patients been investigated for VHL syndrome?
Response 4: We appreciate your important question. The patients included in this study were not investigated for VHL syndrome.
Comments 5: quite a few patients died from non-mRCC related events (18% for >75 yo). What were the causes of death and could they be related to side effects of NIVO+IPI?
Response 5: Thank you for your valuable question. The other causes of death in patients ≥ 75 years are listed below and are not associated with NIVO+IPI treatment. (pneumonia n=3, cerebral hemorrhage n=1, cardiovascular disease n=1 and chronic kidney disease n=1). Therefore, we revised the sentence as follows.
During the median follow-up duration of 21 months (quartile: 8-31 months), among the patients ≥ 75 years old, 14 died of mRCC (42%) and six died of other causes (18%) (pneumonia n=3, cerebral hemorrhage n=1, cardiovascular disease n=1 and chronic kidney disease n=1). (lines 148-151)
Comments 6: In the discussion, I did not find a clear statement that discontinuation of treatment with sunitinib is associated with the acquisition of resistance to this chemotherapeutic and subsequent proliferation of tumor cells ? Could this be the case with NIVO+IPI ? The paper DOI: 10.3892/ijo.2019.4830 may be helpful in this case.
Response 6: We appreciate your valuable comments. While the most of advanced RCCs treated with sunitinib develop intrinsic drug resistance, the main advantages of NIVO+IPI are considered to be the durability of response and long-term survival benefit, which might be a different characteristic from that of sunitinib. Therefore, we revised the sentence in the discussion section as follows.
The main advantages of NIVO+IPI are considered to be the durability of response and long-term survival benefit, which might be a different characteristic from that of sunitinib because it has been reported that most of the advanced mRCCs treated with sunitinib develop intrinsic drug resistance [16]. (lines 203-206)
Reviewer 3 Report
Comments and Suggestions for Authors
With this retrospective multicenter study the authors provide an accurate and valuable account in the treatment of mRCC, especially in elderly patients. Overall, the quality of the article is good. In a contest where the worldwide median age at diagnosis of kidney cancer is nearly 75 years, this study gives new insights to improve the therapeutic management of elderly patients with mRCC. Only few concerns raised from my side to improve the article:
· The title of paragraph 4 should be under the table 3 (lines 171-172)
· Format of table 3 should be the same of table 1 and table 2
Author Response
Comments 1: The title of paragraph 4 should be under the table 3 (lines 171-172)
Response 1: We appreciate your valuable comment. The title of paragraph 4 was moved to the bottom of the table3.
Comments 2: Format of table 3 should be the same of table 1 and table 2
Response 2: Thank you for your important suggestion. The format of Table 3 was revised.